# MONKEYPOX 2022 WITH CROSS INFECTION HYPOTHESIS VIA EPIDEMIOLOGICAL MODEL

## ABSTRACT

A new re-emerging infectious disease of monkeypox 2022 is structurally related to smallpox that is induced by the monkeypox viruses and has caused 59,606 active cases with 18 deaths up to September 15, 2022. To end this ongoing epidemic, there is a need for population-wide control policies like reducing social interaction by keeping social distance, treatment of infected individuals, and restriction on animals, etc. We forecast the progression of the epidemic and come up with an efficient control mechanism by formulating a mathematical model. The biological feasibility and dynamical behavior of the proposed model are then investigated together with sensitivity analysis to obtain the effect of various epidemic parameters mitigating the spread of the disease. Subsequently, by taking non-pharmaceutical and pharmaceutical intervention strategies as control measures, an optimal control theory is applied to mitigate the fatality of the disease to minimize the infectious population and reduce the cost of controls, we construct an objective functional and solve it by using Pontryagin's maximum principle. Finally, extensive numerical simulations are performed to show the impact of the application of intervention mechanisms in controlling the transmission of the monkeypox epidemic.

## 1 INTRODUCTION

The infection of monkeypox is a contagious disease resulting from the orthopoxvirus. This infection is zoonotic and was initially transported to humans by wild rodents in central and western Africa. But human-to-human spread (horizontal transmission) is also possible, particularly within the family home or in the context of care (Farahat et al., 2022). The monkeypox viruses can be diffused by immediate contact with lesions on the skin or mucous membranes of a sick person, as well as by droplets (sneezing, saliva, sputters, etc.) (Singh et al., 2021). Generally, an individual can become infected through contact with patient's environment. It is, therefore, important that patients respect isolation measures throughout the illness. Humans can also become infected through active contact with animals (rodents and monkeys) (Oladoye, 2021). Usually, the monkeypox infection starts from fever, headaches, body aches, weakness, etc. (Deresinski, 2022). The symptoms may lead to the appearance of a blistering rash consisting of fluid-filled blisters that progress to dryness and crusting, then scarring and itching after two days. The bubbles are most concentrated on the face, the forehands, and the feet soles. The disease is more severe in children as well as those who have weak immune systems. Historically, monkeypox was identified first in the 1970s, but recently the re-emerging of the disease, cases are reported in various countries around the globe (ASSESSMENT, 2022). Usually monkeypox virus transmits from human interaction, but there is a significant risk of cross-infection (animal-to-human) spread (Petersen et al., 2019). Therefore, the hypothesis of cross-infection between human and animals play a significant role and can not be neglected.

Modeling and forecasting with the aid of dynamical system is a challenging domain in various discipline, e.g., infectious disease epidemiology (Brauer, 2017; Saravanakumar et al., 2020; Guo et al., 2020), health sciences (Choi et al., 2016), and various other fields of applied science and technology (Rolnick et al., 2022), and therefore attracted the considerable attention of researchers, see for instance, (Das et al., 2020b; Yin et al., 2021; Saha et al., 2021). Similarly, various models demonstrate different outlooks regarding the dynamical behavior of an epidemic (Busenberg & Cooke, 2012; Khajanchi et al., 2018; Das et al., 2020a). With the aim of these mathematical models, researchers want to understand the dynamics of a disease and then suggest control strategies to control or completely eradicate the infection (Chen & Guo, 2016; Kumar et al., 2019). Besides the rich literature on

infectious disease epidemiology, there have been no enough studies found to represent the temporal dynamics of monkeypox 2022, to the best of our knowledge. We try to formulate a model which describes the transmission of monkeypox 2022 to understand the dynamics and suggest a control mechanism with the aid of optimal control theory. We summarize our contributions in this work as follows:

- The cross infection between humans and animals plays a significant role in the dynamics of monkeypox virus transmission. We, therefore, propose a model based on the hypothesis of cross-infection between humans and animals. The model has two blocks: humans and animals.

- The first block describes the evolution of monkeypox in the human population, while the second block represents the evolution of the monkeypox virus among animals.

- Four time-dependent control measures are then introduced in the model to demonstrate the utilization of optimal control measures: to minimize the infectious individuals and maximize the recovered population. Particularly, reducing the risk of disease transmission by educating people to rise awareness of risk factors, treatment of infected individuals, and restrictions on animals.

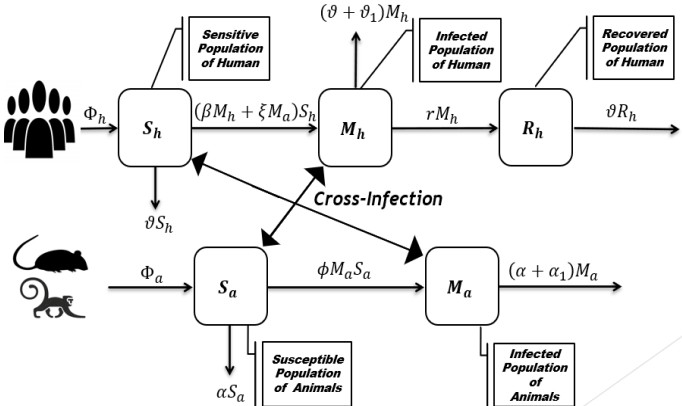

Figure 1: The plot represents the schematic process of the proposed monkeypox virus transmission model (1)

## 2 RELATED WORK

The analysis of infectious diseases with the aid of dynamical systems is a fascinating outlook to predict the dynamics of an epidemic. In the history of infectious disease epidemiology, Kermack and McKendrick were the pioneers to develop the three-population-group epidemiological model (susceptible-infectious-recovered) (Kermack & McKendrick, 1927), where various population groups are employed to signify the infection, demonstrating their progression and interactions. The classical susceptible-infectious-recovered model formulated by Kermack and McKendrick has been modified by incorporating an exposed compartment known as the susceptible-exposed-infectious-recovered model (Anderson & May, 1979), which is also extensively used to delineate the transmission of distinct diseases. Data-driven modeling methods have been also used to investigate the transmission dynamics of infectious diseases (Heesterbeek et al., 2015). It is worthy to mention that the idea of the classical susceptible-infectious-removed model has been further investigated by various researchers to observe the transmission dynamics of distinct epidemics (see for instance Flaxman et al. (2020); Samui et al. (2020); Britton et al. (2020)). Optimal control theory has been extensively used and is very common in infectious disease epidemiology (Rohith & Devika, 2020; Khajanchi et al., 2021). The adjustment of epidemic parameters in a feasible way, by taking the limits on the system to optimize a given function, can be applied with the help of control theory. Both non-pharmaceutical and pharmaceutical control measures can be used to control the infection. Especially, the non-pharmaceutical intervention strategies play a key role. Usually, with the help of optimal control analysis, we are able to know how to eradicate the disease.

A re-emerging infectious disease of monkeypox 2022, was reported in May 2022. Here, we are interested to formulate a model by taking an extended susceptible–infected–recovered-type model with two compartmental blocks: humans and animals. We then discuss the qualitative analysis of the proposed two-strained model. Further, applying the theory of optimal control to understand the progression of the monkeypox virus transmission. Since it is not merely a medical problem, regarding a public health concern, both the combination of non-pharmaceutical and pharmaceutical intervention can be taken into account to propose a control mechanism for the control of monkeypox virus transmission.

## 3 THE MODEL

We propose an epidemiological model for the dynamics of monkeypox based on the cross infection hypothesis: animal to human and human to human. The various compartmental population of the model divided into two blocks: human and animal. The first block represents the evolution of the human population, consequently distributed into three epidemiological groups: sensitive individuals, infected by monkeypox, and recovered individuals, while the second block represents the evolution of animals, divided into two classes: susceptible animals and infected animals. To symbolize the population groups, let us assume that $S_h(t)$ represents the sensitive individuals at time $t$, which are not infected but have a chance to be infected at time $t + \Delta t$ ($\Delta t$ is the small increment in time). $M_h$ denotes the individual infected with monkeypox, and $R_h$ is the recovered individuals. Similarly, the susceptible and infected animals are denoted by $S_a(t)$ and $I_a(t)$, respectively. Due to the assumption of a homogeneously mixed population for the successful transmission of the monkeypox virus, the risky humans will enter the infected human compartment at a rate $\beta$, as well as, the susceptible animal getting infected will move to the infected animal at a rate $\phi$. The individual leaves the infected human class only after they fully recover or die. The recovered individuals will enter $R_h$. Moreover, the complete geometry of the epidemic problem is described by Figure 1, and thus, the evolution of the disease is represented by the following deterministic system of differential equations:

$$
\begin{cases}
\dfrac{dS_h(t)}{dt} = \Phi_h - \beta M_h(t)S_h(t) - \xi M_a(t)S_h(t) - \vartheta S_h(t), \\[2mm]
\dfrac{dM_h(t)}{dt} = \beta M_h(t)S_h(t) + \xi M_a(t)S_h(t) - (\vartheta + \vartheta_1 + r)M_h(t), \quad \dfrac{dR_h(t)}{dt} = rM_h(t) - \vartheta R_h(t), \\[2mm]
\dfrac{dS_a(t)}{dt} = \Phi_a - \phi M_a(t)S_a(t) - \alpha S_a(t), \quad \dfrac{dM_a(t)}{dt} = \phi M_a(t)S_a(t) - (\alpha + \alpha_1)M_a(t),
\end{cases}
\tag{1}
$$

with biologically feasible non-negative initial population sizes

$$
S_h(0) > 0, \quad M_h(0) \geq 0, \quad R_h(0) \geq 0, \quad S_a(0) > 0, \quad M_a(0) \geq 0.
\tag{2}
$$

In the above epidemic problem (1)-(2), the parameters are described as: the newborn of human and animal are denoted by $\Phi_h$ and $\Phi_a$, respectively, while the monkeypox virus transmission rates are $\beta$ and $\xi$. The natural death rate of a human is assumed to be $\vartheta$, while the same ratio for the animal population is denoted by $\alpha$. The monkeypox virus transmission rate from one animal to another is assumed to be $\phi$, and $\alpha_1$ is the death rate that arises from the infection of monkeypox virus in the animal population. Moreover, the disease-induced death rate of a human is represented by $\vartheta_1$, and $r$ is the recovery rate of an infected human.

To proceed, first, we show the mathematical, as well as, the biological feasibility of the proposed epidemic problem. To this end, we show the following result.

**Proposition 1** *The solution of the model (1)-(2) is positive and bounded.*

### 3.1 DYNAMICAL ANALYSIS

In this section, we discuss the temporal dynamics of the model to find the stability conditions for the monkeybox epidemic model. We find the monkeypox-free equilibrium state for the developed model (1) and calculate the reproductive number. Let $W_0$ is the monkeypox-free equilibrium of the model, then, $W_0 = \left(\frac{\Phi_h}{\vartheta}, 0, 0, \frac{\Phi_a}{\alpha}, 0\right)$. Moreover, the reproductive parameter, denoted by $R_o$,

| Parameters | Indices | % Increase or Decrease | Impact on $R_o$ |
|---|---|---|---|
| $\beta$ | 0.3677 | 10 | 3.677 % |
| $\vartheta_1$ | -0.0288 | 10 | 0.288 % |
| $r$ | -0.3316 | 10 | 3.316 % |
| $\phi$ | 0.6322 | 10 | 6.322 % |
| $\alpha_1$ | -0.5099 | 10 | 5.099 % |
| $\alpha$ | -0.7546 | 10 | 7.546 % |

Table 1: Indices of the proposed epidemic problem parameters related with the basic reproductive number, and its relative impact.

represents the average of the secondary infectious produced by an infective whenever put into a sensitive/susceptible individual. To calculate this quantity for the model reported in Eq.(1) by following (Van den Driessche & Watmough, 2002), let us assume that $X = (M_h, M_a)^\top$, then

$$\frac{dX}{dt} = F - V,$$

where, $F$ and $V$ are the 2 by 2 variational matrices at the monkeypox-free equilibrium defined as

$$F = \begin{pmatrix} \frac{\beta \Phi_h}{\vartheta} & \frac{\xi \Phi_a}{\alpha} \\ 0 & \frac{\phi \Phi_a}{\alpha} \end{pmatrix}, \quad V = \begin{pmatrix} \vartheta + \vartheta_1 + r & 0 \\ 0 & \alpha + \alpha_1 \end{pmatrix}.$$

The reproductive number is the spectral radius of the matrix $FV^{-1}$ and takes the following form

$$R_o = R_h + R_a, \quad R_h = \frac{\beta \Phi_h}{\vartheta (\vartheta + \vartheta_1 + r)}, \quad R_a = \frac{\phi \Phi_a}{\alpha (\alpha + \alpha_1)}.$$

Since the reproductive number is the expected average number of secondary infections. It is concluded that whenever $R_o < 1$ the disease will die out, otherwise spread if $R_o > 1$.

### 3.2 BIOLOGICAL INTERPRETATION OF THE BASIC REPRODUCTIVE NUMBER

Since the initial spread of any epidemic is related to the reproductive number, we analyze the normalized sensitivity of the proposed system parameters. The sensitivity of the threshold quantity will enable us to recognize the most sensitive and effective parameters for disease transmission, because a small perturbation in the most sensitive parameter can produce a great influence on the associated epidemic model. To present the prediction for the prevalence of the monkeypox disease, reduction, and persistence in the transmission of infection, we perform sensitivity analysis of the basic reproductive number $R_o$. Let us assume that $\gamma$ is any epidemic parameter, then the normalized forward sensitivity co-efficient (index) related to the basic reproductive number $R_o$ is defined by:

$$\Upsilon_\gamma^{R_o} = \frac{\partial R_o}{\partial \gamma} \times \frac{\gamma}{R_o}.$$

It is clear from the formula of normalized sensitivity coefficient that it may be dependent or independent of the model parameters. We calculate the associated sensitivity indices accordingly listed in Table 1. It can be observed that some of the indices of the model parameters are negative and some are positive. The negative and positive signs demonstrate that the perturbation to these parameters can produce a decrease or increase in the value of the basic reproductive number, respectively. For example, the forward sensitivity index of the parameter $\beta$ is $\Upsilon_\beta^{R_o} = 0.3677$, which indicates that if we increase the value of $\beta$ by 10%, as a result, the value of the basic reproductive number $R_o$ would increase by 3.677%. Similarly, the forward sensitivity indices of $\vartheta_1$ and $\phi$ are 0.0288 and 0.6322, respectively, which implies that if we perturb the value of $\vartheta_1$ and $\phi$ by 10% it would result in increase or decrease in the value of the basic reproductive number by 0.288% and 6.322%, respectively. On the other hand, the sensitivity indices of $r$, $\alpha$, and $\alpha_1$ are negatively associated with the basic reproductive number, i.e., increasing their values would decrease the value of $R_o$. If we increase the value of $r$, $\alpha$, and $\alpha_1$ by 10% it casts the decrease of 15.961% in the value of $R_o$. In this analysis, we observed that the most effective parameters of the proposed epidemic problem are $\beta$, $r$, $\phi$, $\alpha_1$, and $\alpha$, therefore, special attention is required for the parameters with highly sensitive indices to forecast the transmission of monkeypox disease. We now state the dynamics of the model by proving the following results.

**Theorem 1** *If $R_o < 1$, then the dynamical system (1) is locally and globally asymptotically stable around the monkeypox-free equilibrium state of the model and unstable if it is greater than unity.*

## 4    OPTIMAL CONTROL

The application of optimal control theory is one of the important theoretical analyses associated with infectious diseases. We use this tool to produce the proper control mechanism for eliminating infection of monkeypox virus transmission. Our analysis is not limited to the theoretical analysis, and we also perform some numerical experiments to show the effect of the proposed control strategies on the dynamics of monkeypox virus transmission. The key goal is to reduce the infected humans and animals while maximizing the recovered human using the optimal control theory. The parameters with certain assumptions lead to the monkeypox model as described by Eq.(1), which is a coupled system with five state variables $(S_h(t), M_h(t), R_h(t), S_a(t), M_a(t))$. We introduce four control measures $\mu_i(t)$ $(i = 1, 2, \ldots, 4)$ that control the number of risky and infected individuals externally over a given time frame.

### 4.1    REDUCING THE RISK OF HUMAN TO HUMAN AND ZOONOTIC TRANSMISSION

The main prevention measure for monkeypox is educating people to rise awareness of risk factors about the control measures they can take for the reduction of the virus transmission. Due to sufficient information, the population needs to maintain social distance, wear masks, follow the strategy of home isolation, etc. Thus, we introduce the control factors $(1 - \eta_1\mu_1(t))$ and $(1 - \eta_2\mu_2(t))$ to control the interaction between susceptible/risky human $S_h(t)$ with infected human $M_h(t)$ and animal $M_a(t)$, which represent the depletion in $\beta$ and $\xi$, respectively, while $\eta_1$ and $\eta_2$ compute the usefulness of the control measure $\mu_1(t)$ and $\mu_2(t)$ (where $\mu_1(t), \mu_2(t) \in [0, 1]$), respectively. The most successful framework is $\mu_1(t) \equiv 1 \equiv \mu_2(t)$, which indicates that when the interaction of susceptible humans with infected humans and animals is almost perfectly avoided it makes the transmission of the disease to zero. Here, $\mu_1(t) \equiv 1 \equiv \mu_2(t)$ means fully response by implementing the given control mechanism, while $\mu_1(t) \equiv 0 \equiv \mu_2(t)$ implies no response. The intensities of the responses are associated with the behavior of the human population, and so these response intensities are represented by $\mu_1(t)$ and $\mu_2(t)$ as control measures. We maximize the responses using isolation so that they change their behavior and the cost will correspond to a nonlinear function of $\mu_1(t)$ and $\mu_2(t)$. Thus, we wish to find the optimal response for risky individuals with the help of isolation as a control measure.

### 4.2    TREATMENT FOR INFECTED INDIVIDUALS

The treatment of infected individuals not only controls the number of the infected individuals but also influences its development. Although there is no proper treatment for the monkeypox virus infection, smallpox and monkeypox are genetically similar, and there are antiviral drugs against smallpox that can be used for treatment purposes. So in the present scenario, we assume the accessibility of treatment for the infected population. We introduce the term $-\eta_3\mu_3(t)M_h(t)$ as a treatment in the proposed model, where $\eta_3$ represents the treatment rate associated with the intensity $\mu_3(t)$. There are various costs associated with given medication, so we assume that the intensity of treatment control measure $\mu_3(t)$ lies between 0 and unity. The control $\mu_3(t)$ will attempt to change the fraction of the infected population to the recovered population.

### 4.3    RESTRICTION ON ANIMALS

While it seems to be difficult that how to restrict animals from the transmission of the monkeypox infection, it is possible. In the current situation, various countries have restricted the importation of animals (rodents) and non-human primates. Animals that are infected with monkeypox should be placed into quarantine immediately to be isolated from other animals. Moreover, an animal that has close contact with another infected animal should be also isolated and accordingly quarantined to observe the symptoms of monkeypox for 30 days. We introduce the control factor $(1 - \eta_4\mu_4(t))$ to control the interactions of susceptible and infected animals, where $\mu_4(t) \in [0, 1]$.

In this section, the main objective is to obtain the optimal control strategy that minimizes the infected population with the aid of the above control measures and with the minimum associated cost. Thus, the admissible set of control measures $\mu_i(t)$ is defined by

$$U = \{\mu_i(t), i = 1, 2, \ldots, 4 : \quad 0 \le \mu_i(t) \le 1, \quad t \in [0, T]\}.$$

We, therefore, develop the control problem by keeping in view the above strategies with the objective functional $W(\{\mu_i\})$ to be minimized:

$$W(\mu_i(t), i = 1, \ldots, 4) = \int_0^T h_1 M_h(t) dt + \frac{1}{2} \int_0^T \sum_{i=1}^4 \kappa_i \mu_i^2(t) dt, \tag{3}$$

subject to

$$\frac{dS_h(t)}{dt} = \Phi_h - \beta \{1 - \eta_1 \mu_1(t)\} M_h(t) S_h(t) - \xi \{1 - \eta_2 \mu_2(t)\} M_a(t) S_h(t) - \vartheta S_h(t),$$

$$\frac{dM_h(t)}{dt} = \beta \{1 - \eta_1 \mu_1(t)\} M_h(t) S_h(t) + \xi \{1 - \eta_2 \mu_2(t)\} M_a(t) S_h(t)$$
$$- \{\vartheta + \vartheta_1 + r + \eta_3 \mu_3(t)\} M_h(t),$$

$$\frac{dR_h(t)}{dt} = \{r + \eta_3 \mu_3(t)\} M_h(t) - \vartheta R_h(t), \frac{dS_a(t)}{dt} = \Phi_a - \phi \{1 - \eta_4 \mu_4(t)\} M_a(t) S_a(t) - \alpha S_a(t),$$

$$\frac{dM_a(t)}{dt} = \phi \{1 - \eta_4 \mu_4(t)\} M_a(t) S_a(t) - \{\alpha + \alpha_1\} M_a(t),$$

$$\tag{4}$$

with the initial population sizes in Eq. (2). In Eq. (3), the integrand represents the value of cost at time $t$, while the function $W$ shows the sum of the cost described by the integrand or the total incurred cost. The parameters $h_1$ and $\kappa_i$'s are non-negative parameters that are weight constants to balance the units of the integrand. The control measures $\mu_i^*$ ($i = 1, 2, 3, 4$) exist in the admissible control set $U$ that minimize $W$. We now discuss the existence of optimal control for our proposed control problem (4), then use the well-known Pontryagin's maximum principle for characterization and getting the necessary conditions of the optimal controls. The following result will be presented to ensure the existence of $\mu_i^*$ that minimizes the function $W$.

**Theorem 2** *There exist optimal controls $\mu^*(t) = (\mu_1(t), \mu_2(t), \mu_3(t), \mu_4(t))$ in $U$ that minimize the objective function $W$ associated with the control problem in Eqs.(4)–(3).*

Since the above result ensures the existence of the controls to minimize the objective functional (3) subject to the state system (4), we then derive the necessary conditions for characterization of the optimal control problem using Pontryagin's maximum principle, see Theorem 3 in the appendix.

## 5 NUMERICAL EXPERIMENTS

We perform numerical experiments to test the model predictions and verify the analytical findings. We utilize a well-known numerical procedure of the Runge-Kutta method of the 4th order. First, we discretize the model and develop the algorithm to perform the numerical simulations.

### 5.1 DISCRITIZATION

To discritize the model, we set

$$X = \begin{pmatrix} S_h \\ M_h \\ R_h \\ S_a \\ M_a \end{pmatrix}, \quad F = \begin{pmatrix} \Phi_h - \beta M_h S_h - \xi M_a S_h - \vartheta S_h \\ \beta M_h S_h + \xi M_a S_h - (\vartheta + \vartheta_1 + r) M_h \\ r M_h - \vartheta R_h \\ \Phi_a - \phi M_a S_a - \alpha S_a \\ \phi M_a S_a - (\alpha + \alpha_1) M_a \end{pmatrix},$$

$$Y = \begin{pmatrix} \varphi_1 \\ \varphi_2 \\ \varphi_3 \\ \varphi_4 \\ \varphi_5 \end{pmatrix}, \quad G = \begin{pmatrix} \Phi_h - \beta \{1 - \eta_1 \mu_1\} M_h S_h - \xi \{1 - \eta_2 \mu_2\} M_a S_h - \vartheta S_h \\ \beta \{1 - \eta_1 \mu_1\} M_h S_h + \xi \{1 - \eta_2 \mu_2\} M_a S_h - \{\vartheta + \vartheta_1 + r + \eta_3 \mu_3\} M_h \\ \{r + \eta_3 \mu_3\} M_h - \vartheta R_h \\ \Phi_a - \phi \{1 - \eta_4 \mu_4\} M_a S_a - \alpha S_a \\ \phi \{1 - \eta_4 \mu_4\} M_a S_a - \{\alpha + \alpha_1\} M_a \end{pmatrix},$$

and

$$H = \begin{pmatrix} \{\varphi_1 - \varphi_2\}\{\beta(1 - \eta_1\mu_1^*)M_h^* + \xi(1 - \eta_2\mu_3^*)M_a\} + \varphi_1^*\vartheta, \\ \{\varphi_1 - \varphi_2\}\{\beta(1 - \eta_1\mu_1^*)S_h^*\} - \{\vartheta + \vartheta_1 + r + \eta_3\mu_3^*\}\varphi_2 - \{r + \mu_3^*\}\varphi_3 - h_1, \\ \vartheta\varphi_3, \{\varphi_4 - \varphi_5\}\{\phi(1 - \eta_4\mu_4^*)M_a\} - \alpha\varphi_4, \\ \{\varphi_1 - \varphi_2\}\{\xi(1 - \eta_2\mu_2^*)S_h^*\} + \{\alpha + \alpha_1\}\varphi_5 + \{\varphi_4 - \varphi_5\}\{\phi(1 - \eta_4\mu_4^*)S_a^*\}, \end{pmatrix},$$

then Eq.(1), Eq.4), and Eq.(6) can be re-casted as

$$\frac{dX(t)}{dt} = F(t, X(t)), \quad \frac{dX(t,\mu)}{dt} = G(t, X(\mu)), \quad \frac{dY(t)}{dt} = H(t, \varphi). \tag{5}$$

The application of forward and backward Runge-Kutta method of order four gives

$$X_{i+1} = X_i + \frac{l}{6}(h_1 + 2h_2 + 2h_3 + h_4), \quad Y_{i-1} = Y_i - \frac{l}{6}(k_1 + 2k_2 + 2k_3 + k_4), \tag{6}$$

where

$$h_1 = F(t_n, X_n), \quad h_2 = F\left(t_n + \frac{l}{2}, X_n + \frac{lh_1}{2}\right), \quad h_3 = F\left(t_n + \frac{l}{2}, X_n + \frac{lh_2}{2}\right),$$

$$h_4 = F(t_n + h, X_n + lh_3), \quad k_1 = F(t_n, X_n), \quad k_2 = F\left(t_n - \frac{l}{2}, X_n - \frac{lk_1}{2}\right),$$

$$k_3 = F\left(t_n - \frac{l}{2}, X_n - \frac{lk_2}{2}\right), \quad k_4 = F(t_n - k, X_n - lk_3).$$

Thus the rest of algorithms can be concluded as:

---

**Algorithm 1** Runge-Kutta Method (RK4)

---

1: **Input:** Endpoints $t_0$, $t_{\max}$, integer $n$, parametric values, initial conditions
2: **Output:** approximation $S_h$, $M_h$, $R_h$, $S_a$, $M_a$ at $(n+1)$ values of $t$
3: **Parameters and Initial Conditions:** Setting the values of epidemic parameters and initial sizes for compartmental populations
4: **for** $i = 1, \cdots, n$ **do**
5:     **Recursive Formula:** $X_{i+1}$ for both control and without control system as given in Eq.(5) and Eq.(6)
6: **end for**
7: **for** $i = 1, \cdots, n, j = n + 2 - i$ **do**
8:     **Recursive Formula:** $Y_{i-1}$ as given in Eq.(5) and Eq.(6)
9: **end for**
10: **Optimal Control:** Plugging optimal control variables as given by Eq.(7)
11: **Output:** $\left(t_i, S_h^{i+1}, M_h^{i+1}, R_h^{i+1}, S_a^{i+1}, M_a^{i+1}\right)$

---

## 5.2 DISCUSSION

We perform numerical simulations to discuss sensitivity analysis and the application of optimal control strategies. We conduct numerical experiments to present the validation of our theoretical findings for the model parameters and initial sizes of populations as specified in Table 2. It could be noted that some of the parameters are directly correlated to the basic reproductive number, $R_o$, while some are negatively correlated, as shown in Figure 2. Figure 3 represents the contour plot that describes the dependency of the basic reproductive number, $R_o$, on $\beta$ (disease transmission co-efficient of humans) and $\phi$ (disease transmission co-efficient of animals); $\beta$ and $\vartheta$ (natural death rate); $\beta$ and $r$ (recovery rate of infected population); and $\phi$ and $\alpha_1$ (the death rate arises from monkeypox in animal population). It is very much clear from Table 1 and Figure 2 that the epidemic parameters, namely, $\beta$ and $\phi$ have positive indices, and are also associated with the susceptible population. If we increase the value of $\beta$ and $\phi$, it will increase the value of basic reproductive number $R_o$ and cross the value $R_o = 1$, which leads to a substantial outbreak of the monkeypox virus transmission. To maintain the value of the basic reproductive number and deduce a favorable

control method to eradicate the spread of monkeypox virus transmission, we need to restrict the value of these parameters, which is possible with the help of contact tracing and maintaining social distancing. On the other hand, the parameters with negative indices are $\vartheta$, $r$, $\alpha$, and $\alpha_1$ as shown in Figure 2, while the relative effect is reported in Figure 3. Moreover, whenever the value of those parameters having negative indices increases, the basic reproductive number $R_o$ will decrease, and if its value becomes less than unity, the infection will no longer persist. So, treatment of infected humans through medication has been incorporated with the aid of optimal control to eliminate the monkeypox virus. To observe the influence of intervention strategies on the transmission dynamics of monkeypox, we perform the numerical simulations of the proposed optimal control problem with the help of the Runge-Kutta (RK4) scheme, as concluded in Algorithm 1. Moreover, the time frame is taken to be 20 units, and the value of parameters are borrowed from Table 2 while investigating and implementing the optimal control mechanism. To investigate the effect of optimal control measures for the monkeypox virus transmission, we execute the proposed problem in two folds: without control $\mu_1(t) = \mu_2(t) = \mu_3(t) = \mu_4(t) = 0$ and with the combination of four controls $(\mu_1(t), \mu_2(t), \mu_3(t), \mu_4(t))$ which leads to the results as presented in Figure 4. These graphs respectively demonstrate the dynamics of human and animal compartments of the proposed model under no control and with controls. The black dashed and red dashed curves respectively represent the dynamics of each compartmental population with and without the utilization of control strategies to highlight the effect of optimal policies implementation, see Figure 4. A significant reduction in the infected population, as well as, an increase in the non-infected population can be seen with the application of optimal control measures, whenever, compared without intervention strategies. We conclude that the combination of suggested optimal control measures can achieve a significant reduction in monkeypox virus transmission whenever applied in a true sense.

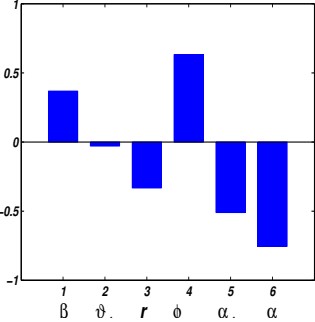

Figure 2: The plot represents the sensitivity indices and most sensitive epidemic parameters related to basic reproductive number $R_o$.

## 6  CONCLUSION

To keep the current scenario of the monkeypox virus transmission in our mind, we investigated and proposed a model for the dynamics of monkeypox virus transmission. Both theoretical and numerical analyses of the proposed epidemic problem have been studied with the aid of stability theory. Positivity, as well as, the boundedness of the states of the epidemic problem guarantees that the considered model is a well-defined dynamical system. We performed the local and global dynamics of the model and derived the stability conditions. The proposed model has several epidemic parameters and so with the help of normalized sensitivity analysis, the most significant parameters are quantified. It could be observed that disease transmission from both human-to-human and animals to human play a significant role in disease transmission. Besides the disease transmission rate, the parameter $r$ is also very effective and has a major effect on infected individuals and $R_o$. Moreover, we modified the proposed model by incorporating optimal measures with the aid of control theory to mitigate the monkeypox disease burden and describe the effect of control policies implementation, and as a result to stop the disease from spreading. The findings of our optimal control problem show that the maintenance of the social distancing, tracing contact, transportation of animals and treatment of infected individuals may support mitigating monkeypox virus spreading in the current scenario of the epidemic.

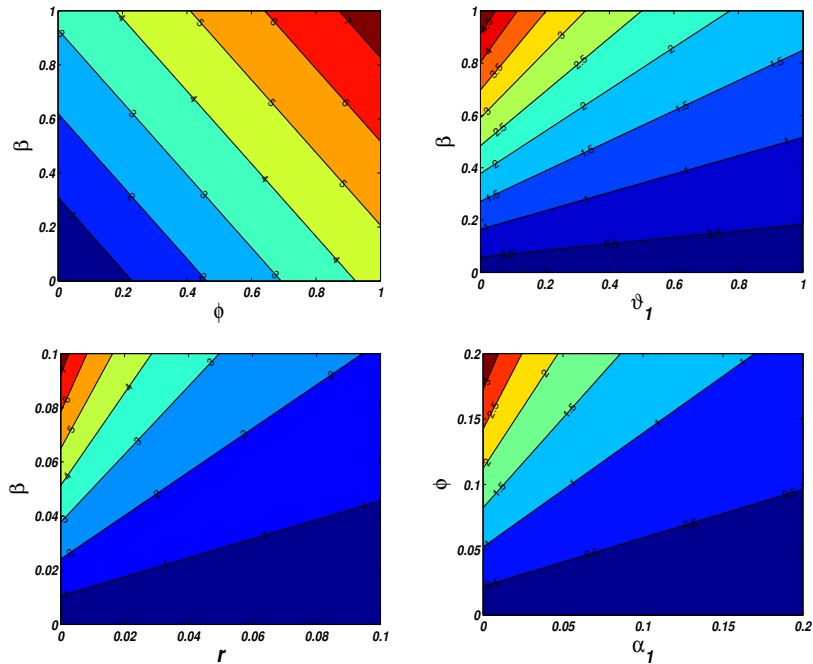

Figure 3: Contour plots of the basic reproductive number $R_o$ to show the sensitivity analysis. The basic reproductive number is taken to be the function of $\beta$, $\phi$ and $\vartheta_1$ to show the relative effect of these parameters on the basic reproductive number.

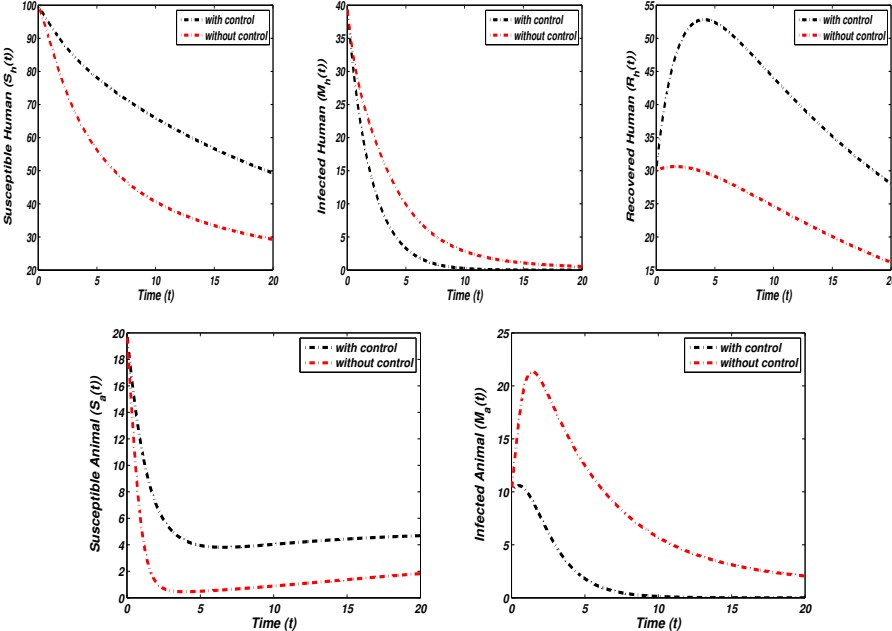

Figure 4: The plots represent the dynamics of the compartmental population with the implementation and without the implementation of control strategies

Given the increased development of fractional calculus, many operators of fractional orders were introduced to capture more valuable information. In our future work, we will generalize the proposed model to its associated fractional version to discuss the dynamics of monkeypox virus disease.

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

## APPENDIX

### A.1 PROOF OF PROPOSITION 1

*Proof.* The differentiability of the right-hand side of the proposed model equations implies the existence of a unique maximal solution for any associated Cauchy problem. So, the solution of the first equation of the model looks like

$$S_h(t) = \exp\left\{-\vartheta t - \int_0^t (\beta M_h(x) + \xi M_a(x))\, dx\right\}$$
$$\times \Phi_h t \exp\left\{\vartheta t + \int_0^x (\beta M_h(y) + \xi M_a(y))\, dy\right\} dx$$
$$+ S_h(0) \exp\left\{-\vartheta t - \int_0^t (\beta M_h(x) + \xi M_a(x))\, dx\right\} > 0.$$

The solution of second equation of the model (1) takes the form

$$M_h(t) = \exp\left\{-(\vartheta + \vartheta_1 + r)t + \int_0^t \beta S_h(x)dx\right\} \int_0^t \xi M_a(x) S_h(x)$$
$$\times \exp\left\{(\vartheta + \vartheta_1 + r)x + \int_0^x \beta S_h(y)dy\right\} dx$$
$$+ M_h(0) \exp\left\{-(\vartheta + \vartheta_1 + r)t + \int_0^t \beta S_h(x)dx\right\} \geq 0.$$

Clearly, we see that $S_h(t) > 0$ and $M_h(t) \geq 0$ for $S_h(0) > 0$ and $M_h(0) \geq 0$. Following the same steps, we can easily show that $R_h(t) \geq 0$, $S_a(t) > 0$ and $M_a(t) \geq 0$. Moreover, to show the boundedness, let us assume that the total human and animal populations are denoted by $N_h(t)$ and $N_a(t)$ respectively, then the total dynamics of humans and animal looks like

$$\frac{dN_h(t)}{dt} \leq \Phi_h - \vartheta N_h(t), \quad \frac{dN_a(t)}{dt} \leq \Phi_a - \alpha N_a(t).$$

It is easy to solve the above inequalities and then taking unbounded limit i.e., $t$ approaches to infinity, $N_h(t) \to \frac{\Phi_h}{\vartheta}$ and $N_a(t) \to \frac{\Phi_a}{\alpha}$, and thus the feasible region for the proposed model is given by

$$\Omega = \left\{ (S_h, M_h, R_h, S_a, M_a) \in R_+^5, \quad N_h \leq \frac{\Phi_h}{\vartheta}, \quad N_a \leq \frac{\Phi_a}{\alpha} \right\}.$$

A.2 PROOF OF THEOREM 1

*Proof.* We use the approach of linear stability analysis and linearize the proposed model around the Monkeypox free equilibrium, $W_0$, then the variational matrix may takes the form

$$J(W_0) = \begin{pmatrix} -\vartheta & -\frac{\beta \Phi_h}{\vartheta} & 0 & 0 & -\frac{\xi \Phi_a}{\alpha} \\ 0 & \frac{\beta \Phi_h}{\vartheta} - (\vartheta + \vartheta_1 + r) & 0 & 0 & \frac{\xi \Phi_a}{\alpha} \\ 0 & r & -\vartheta & 0 & 0 \\ 0 & 0 & 0 & -\alpha & -\frac{\phi \Phi_a}{\alpha} \\ 0 & 0 & 0 & 0 & -(\alpha + \alpha_1) \end{pmatrix}.$$

Let the associated eigenvalues of $J(W_0)$ are $\lambda_i$, where, $i = 1, 2, \ldots, 5$, then we obtain

$$\lambda_1 = -\vartheta = \lambda_3, \lambda_2 = -(\vartheta + \vartheta_1 + r)(1 - R_h), \lambda_4 = -\alpha, \lambda_5 = -(\alpha + \alpha_1).$$

Obviously, $\lambda_1$, $\lambda_3$, $\lambda_4$ and $\lambda_5$ are negative while $\lambda_2$ is negative whenever $R_h < 1$ holds. Since, $R_h < 1$ provided that $R_0 < 1$, therefore we conclude that the proposed Monkeypox model is locally asymptotically stable at the disease-free state $W_0$ subject to the condition $R_0 < 1$ holds. To discuss the global dynamics, let us define $S_h^o = \frac{\Phi_h}{\vartheta}$ and $S_a^o = \frac{\Phi_a}{\alpha}$, and a real-valued function, $H(t) = H_0 + H_1$, such that

$$H_0 = \left\{ S_h - S_h^0 \right\} + \left\{ S_a - S_a^0 \right\}, \quad H_1 = M_h + R_h + M_a.$$

Clearly, $H(t)$ is non-negative and its temporal differentiation with the application of some algebraic manipulation leads to the assertion

$$\frac{dH}{dt} = -\vartheta \left\{ S_h - S_h^o \right\} - \alpha \left\{ S_a - S_a^o \right\} - (\vartheta + \vartheta_1) M_h - \vartheta R_h - (\alpha + \alpha_1) M_a,$$

implies that $\frac{dH}{dt} < 0$. Also it could be noted that, whenever, $S_h = S_h^o$ and $S_a = S_a^o$, then $\frac{dH}{dt} = 0$, which prove that $H(t)$ is Lyapunov function and the proposed model is stable globally asymptotically.

A.3 PROOF OF THEOREM 2

*Proof.* Since the solutions of model (1) are bounded and non-negative for the initial population sizes (2). The objective functional is also non- negative which ultimately implies that $W$ is bounded. Thus to minimize the sequence of controls $\mu^\tau(t) = (\mu_1^\tau(t), \mu_2^\tau(t), \mu_3^\tau(t), \mu_4^\tau(t)) \in U$ exists, such that

$$\lim_{\tau \to \infty} W(\mu^\tau(t)) = \inf_{\mu \in U} W(\mu(t)).$$

The controls in set $U$ are bounded uniformly in $L^\infty$ space which implies that they are bounded uniformly in $L^2([0, T])$ space. Since, $L^2$ space is reflexive, so there exists a subsequence $\mu^*(t) \in U$ such that

$$\mu_1^\tau(t) \to \mu_1^*(t), \quad \mu_2^\tau(t) \to \mu_2^*(t), \quad \mu_3^\tau(t) \to \mu_3^*(t), \quad \mu_4^\tau(t) \to \mu_4^*(t)$$

weakly in $L^2$ space as $\tau$ approaches $\infty$, then the sequence defined by $z^\tau = (S_h^\tau, M_h^\tau, R_h^\tau, S_a^\tau, M_a^\tau)$ is bounded uniformly corresponding to $\mu^\tau(t)$. It can be observed from the right-hand side of model (1) that they are uniformly bounded which gives the equi-continuity of the state sequence and uniform

bounded-ness of the derivatives for $z^\tau$. The *Arzel–Ascoli* theorem then gives that there exist $z^\tau = (S_h^\tau, M_h^\tau, R_h^\tau, S_a^\tau, M_a^\tau)$ such that $z^\tau \to z^*$ uniformly on $[0, T]$.

Further, to complete the proof, let us assume a proper subsequence passing through the limit of the system (1) and corresponding to $\mu_1^*(t), \mu_2^*(t), \mu_3^*(t), \mu_4^*(t)$, we can obtain the optimal solution $z^*$. Thus, the lower semi-continuity of $L^2$ norm with respect to $L^2$ weak convergence gives that

$$\inf_{\mu \in U} W(\mu^\tau) = \lim_{\tau \to \infty} W(\mu^\tau) \geq \int_0^T h_1 M_h dt + \frac{1}{2} \int_0^T \sum_{i=1}^4 \kappa_i \mu_i^2 dt = W(\mu^*).$$

Therefore, $\mu^*(t) = (\mu_1^*(t), \mu_2^*(t), \mu_3^*(t), \mu_4^*(t))$ is an optimal control measure.

## A.4 Characterization of Optimal Control

**Theorem 3** *There exists optimal controls $\mu^* = (\mu_1, \mu_2, \mu_3, \mu_4)$ in $U$ and the corresponding solutions of optimal state $z^* = (S_h^*, M_h^*, R_h^*, S_a^*, M_a^*)$ of the Eq.(4) that minimize $W$ (3), then the adjoint variables $\varphi_1$, $\varphi_2$, $\varphi_3$, $\varphi_4$, and $\varphi_5$ satisfying the equations:*

$$\varphi_1'(t) = \{\varphi_1 - \varphi_2\} \{\beta (1 - \eta_1 \mu_1) M_h + \xi(1 - \eta_2 \mu_2) M_a\} + \varphi_1 \vartheta,$$

$$\varphi_2'(t) = \{\varphi_1 - \varphi_2\} \{\beta (1 - \eta_1 \mu_1) S_h\} + \{\vartheta + \vartheta_1 + r + \eta_3 \mu_3\} \varphi_2 - \{r + \mu_3\} \varphi_3 - h_1,$$

$$\varphi_3'(t) = \vartheta \varphi_3, \quad \varphi_4'(t) = \{\varphi_4 - \varphi_5\} \{\phi (1 - \eta_4 \mu_4) M_a\} + \alpha \varphi_4,$$

$$\varphi_5'(t) = \{\varphi_1 - \varphi_2\} \{\xi (1 - \eta_2 \mu_2) S_h\} + \{\alpha + \alpha_1\} \varphi_5 + \{\varphi_4 - \varphi_5\} \{\phi (1 - \eta_4 \mu_4) S_a\},$$

*with the transversal condition:*

$$\varphi_1(T) = \varphi_2(T) = \varphi_3(T) = \varphi_4(T) = \varphi_5(T) = 0.$$

*Further, the associated optimal measures $\mu^*(t) = (\mu_1^*(t), \mu_2^*(t), \mu_3^*(t), \mu_4^*(t))$ are:*

$$\mu_1^* = \frac{1}{\kappa_1} \{\varphi_2 - \varphi_1\} \beta \eta_1 M_h S_h, \quad \mu_2^* = \frac{1}{\kappa_2} \{\varphi_1 - \varphi_2\} \xi \eta_2 M_a S_h, \quad \mu_3^* = \frac{1}{\kappa_3} \{\varphi_2 - \varphi_3\} \eta_3 M_h,$$

$$\mu_4^* = \frac{1}{\kappa_4} \{\varphi_5 - \varphi_4\} \eta_4 M_a S_a.$$

*Proof.* In order to investigate the necessary conditions for the control measures using the Pontryagin'smaximum principle for the proposed model (4), we define the Hamiltonian $H$ for every $t \in [0, T]$ as:

$$H = h_1 M_h(t) dt + \frac{1}{2} \sum_{i=1}^4 \kappa_i \mu_i^2(t) + \varphi(t) g(z, \mu),$$

where $\varphi(t) = (\varphi_1(t), \varphi_2(t), \varphi_3(t), \varphi_4(t), \varphi_5(t), )$, $g = (g_1, g_2, g_3, g_4, g_5)$, and $g_1, g_2, g_3, g_4$ and $g_5$ respectively represent the right hand side of the first, second, third, fourth and fifth equation of the system (4). We derive the minimized Hamiltonian with the help of Pontryagin's maximum principle which minimize the objective function $W$ (3). The Pontryagin's maximum principle relates the state variables of the model and the objective functional with including adjoint variables. Therefore, by using the Pontryagin's principle there exists $\varphi_1'(t)$, $\varphi_2'(t)$, $\varphi_3'(t)$, $\varphi_4'(t)$ and $\varphi_5'(t)$ that satisfies

$$\varphi_1'(t) = \frac{\partial H}{\partial S_h}, \varphi_2'(t) = \frac{\partial H}{\partial M_h}, \varphi_3'(t) = \frac{\partial H}{\partial R_h}, \varphi_4'(t) = \frac{\partial H}{\partial S_a}, \varphi_5'(t) = \frac{\partial H}{\partial M_a},$$

with the transversal conditions $\varphi_1(T) = \varphi_2(T) = \varphi_3(T) = \varphi_4(T) = \varphi_5(T) = 0$. Now by using the characterization of controls we have

$$\frac{\partial H}{\partial \mu_1(t)} = 0, \quad \frac{\partial H}{\partial \mu_2(t)} = 0, \quad \frac{\partial H}{\partial \mu_3(t)} = 0, \quad \frac{\partial H}{\partial \mu_4(t)} = 0.$$

From $\frac{\partial H}{\partial \mu_i} = 0$ ($i = 1, 2, 3, 4$), and at $\mu_1(t) = \mu_1^*(t)$, $\mu_2(t) = \mu_2^*(t)$, $\mu_3(t) = \mu_3^*(t)$ and $\mu_4(t) = \mu_4^*(t)$ we obtain

$$\mu_1^* = \frac{1}{\kappa_1} \{\varphi_2 - \varphi_1\} \beta \eta_1 M_h S_h, \quad \mu_2^* = \frac{1}{\kappa_2} \{\varphi_1 - \varphi_2\} \xi \eta_2 M_a S_h, \mu_3^* = \frac{1}{\kappa_3} \{\varphi_2 - \varphi_3\} \eta_3 M_h,$$

$$\mu_4^* = \frac{1}{\kappa_4} \{\varphi_5 - \varphi_4\} \eta_4 M_a S_a.$$

| Parameter | Value | Parameter | Value | Parameter | Value |
|-----------|-------|-----------|-------|-----------|-------|
| $\Phi_h$ | 0.9430 | $\beta$ | 0.0003 | $\xi$ | 0.0044 |
| $\vartheta$ | 0.4570 | $\vartheta_1$ | 0.0910 | $r$ | 0.0590 |
| $\Phi_a$ | 0.6000 | $\phi$ | 0.0790 | $\alpha$ | 0.1200 |
| $\alpha_1$ | 0.0900 | $\eta_1$ | 0.0900 | $\eta_2$ | 0.5000 |
| $\eta_3$ | 0.0400 | $\eta_4$ | 0.0400 | $\eta_5$ | 0.0500 |
| $h_1$ | 0.6000 | $h_2$ | 0.9000 | $k_1$ | 0.4400 |
| $k_2$ | 0.0200 | $k_3$ | 0.2000 | $k_4$ | 0.2000 |
| $k_5$ | 0.2000 | | | | |

Table 2: Parametric values used in the numerical simulation of the optimal control problem

By considering the lower and upper bound for $\mu_1(t)$, $\mu_2(t)$, $\mu_3(t)$, and $\mu_4(t)$, we leads to the following assertions:

$$\mu_1^*(t) = \min\left\{\max\left\{0, \frac{1}{\kappa_1}\left\{\varphi_2(t) - \varphi_1(t)\right\}\beta\eta_1 M_h(t)S_h(t)\right\}, 1\right\},$$

$$\mu_2^*(t) = \min\left\{\max\left\{0, \frac{1}{\kappa_2}\left\{\varphi_1(t) - \varphi_2(t)\right\}\xi\eta_2 M_a(t)S_h(t)\right\}, 1\right\},$$

$$\mu_3^*(t) = \min\left\{\max\left\{0, \frac{1}{\kappa_3}\left\{\varphi_2(t) - \varphi_3(t)\right\}\eta_3 M_h(t)\right\}, 1\right\},$$
(7)

$$\mu_4^*(t) = \min\left\{\max\left\{0, \frac{1}{\kappa_4}\left\{\varphi_5(t) - \varphi_4(t)\right\}\eta_4 M_a(t)S_a(t)\right\}, 1\right\}.$$

It is also clear from the Hamiltonian $H$ that

$$\frac{\partial^2 H}{\partial\mu_1^2} = \kappa_1^2 > 0, \quad \frac{\partial^2 H}{\partial\mu_2^2} = \kappa_2^2 > 0, \quad \frac{\partial^2 H}{\partial\mu_3^2} = \kappa_3^2 > 0, \quad \frac{\partial^2 H}{\partial\mu_4^2} = \kappa_4^2 > 0,$$

which implies that the optimal measures minimize $(H)$.

We state the optimality system with the help of $\mu_1^*$, $\mu_2^*$, $\mu_3^*$ and $\mu_4^*$. Thus, the optimality system that minimize $H$ at $(S_h^*, M_h^*, R_h^*, S_a^*, M_a^*, \mu_1^*, \mu_2^*, \mu_3^*, \mu_4^*, \varphi_1, \varphi_2, \varphi_3, \varphi_4)$ becomes:

$$\frac{dS_h}{dt} = \Phi_h - \beta\left\{1 - \eta_1\mu_1^*\right\}M_h^*S_h^* - \xi\left\{1 - \eta_2\mu_2^*\right\}M_a^*S_h^* - \vartheta S_h^*,$$

$$\frac{dM_h}{dt} = \beta\left\{1 - \eta_1\mu_1^*\right\}M_h^*S_h^* + \xi\left\{1 - \eta_2\mu_2^*\right\}M_a^*S_h^* - \left\{\vartheta + \vartheta_1 + r + \eta_3\mu_3^*\right\}M_h^*,$$

$$\frac{dR_h}{dt} = \left\{r + \mu_3(t)\right\}M_h^* - \vartheta R_h^*, \quad \frac{dS_a}{dt} = \Phi_a - \phi\left\{1 - \eta_4\mu_4(t)\right\}M_a^*S_a^* - \alpha S_a^*,$$

$$\frac{dM_a}{dt} = \phi\left\{1 - \eta_4\mu_4^*\right\}M_a^*S_a^* - \left\{\alpha + \alpha_1\right\}M_a^*,$$

with initial population sizes $S_h^*(0) > 0$, $M_h^*(0) \geq 0$, $R_h^*(0) \geq 0$, $S_a^*(0) \geq 0$, $M_a^*(0) \geq 0$, and the associated co-states (adjoint variables) are

$$\varphi_1^{'} = \left\{\varphi_1 - \varphi_2\right\}\left\{\beta\left(1 - \eta_1\mu_1^*\right)M_h^* + \xi(1 - \eta_2\mu_3^*)M_a\right\} + \varphi_1^*\vartheta,$$

$$\varphi_2^{'} = \left\{\varphi_1 - \varphi_2\right\}\left\{\beta\left(1 - \eta_1\mu_1^*\right)S_h^*\right\} - \left\{\vartheta + \vartheta_1 + r + \eta_3\mu_3^*\right\}\varphi_2 - \left\{r + \mu_3^*\right\}\varphi_3 - h_1,$$

$$\varphi_3^{'} = \vartheta\varphi_3, \quad \varphi_4^{'} = \left\{\varphi_4 - \varphi_5\right\}\left\{\phi\left(1 - \eta_4\mu_4^*\right)M_a\right\} - \alpha\varphi_4,$$

$$\varphi_5^{'} = \left\{\varphi_1 - \varphi_2\right\}\left\{\xi\left(1 - \eta_2\mu_2^*\right)S_h^*\right\} + \left\{\alpha + \alpha_1\right\}\varphi_5 + \left\{\varphi_4 - \varphi_5\right\}\left\{\phi\left(1 - \eta_4\mu_4^*\right)S_a^*\right\},$$

where $\varphi_1(T) = \varphi_2(T) = \varphi_3(T) = \varphi_4(T) = \varphi_5(T) = 0$ are the transversal conditions, and the four control variables are stated by Eq.(7). Hence, the proof.

