# OpenReview forum: "Monkeypox with Cross Infection Hypothesis via Epidemiological Mode"
_ICLR.cc/2023/Conference — Submitted to ICLR 2023_

### Official Review · Reviewer_8Pvu · 2022-10-19

**Confidence:** 5
**Correctness:** 4
**Technical Novelty And Significance:** 2
**Empirical Novelty And Significance:** 2
**Recommendation:** 3

**Clarity, Quality, Novelty And Reproducibility:**

The clarity is good (although I would suggest removing the double arrow heads on the cross-infection figure), quality is high for a paper in its field, novelty is modest and reproducibility would be expected to be high.


**Strength And Weaknesses:**

Considered across the spectrum of modelling approaches for all diseases, a two population compartmental model is at the lower level of complexity of those in use; its formal analysis by optimal control theory is less common and would be of interest to numerical modellers working in this space.  The investigation here is of a rather theoretical nature, more so than practical: that is, the problem is formulated in a general sense, rather than in collaboration with an actual disease control program or with a confrontation against real world data.  This would be suitable for a mathematical disease modelling journal; however, I do not see it as being suitable for ICLR as I cannot discern a novel contribution in the domain of deep learning (or an adjacent field).


**Summary Of The Paper:**

In their manuscript entitled, "Monkeypox 2022 with cross infection hypothesis via epidemiological model", the authors present a two-population (human and animal) compartmental model for describing the evolution of the 2022 monkeypox outbreak.  Given various intervention options the authors present an investigation of optimal control policies for this system.


**Summary Of The Review:**

An interesting paper for a mathematical disease modelling audience but not for ICLR.

---

### Official Review · Reviewer_8Yf3 · 2022-11-01

**Confidence:** 3
**Correctness:** 2
**Technical Novelty And Significance:** 2
**Empirical Novelty And Significance:** 1
**Recommendation:** 3

**Clarity, Quality, Novelty And Reproducibility:**

Novelty

As mentioned above, the main novelty of the work lies in explicitly modeling the human and animal interaction; however, other studies have also incorporated such spillover modeling (see, e.g., the review by [Lloyd-Smith et al., Science 2009] for references). The remaining derivations and methodological approaches appear to be standard methods used in expected ways.

Quality

I am not an expert in the area, and I did not verify the derivations in detail. Nevertheless, they appeared generally correct.

The experiments and related discussion was rather limited. Of course, the main contribution of this work is the model itself. Still, there was no type of data-driven evaluation of even qualitative discussion about whether the results from the model were plausible. Publicly-available data for at least the human side of such models seem to be available (e.g., the data used in [Purkayastha et al., BMC Infectious Disease 2021], but many others). While the exact numbers for the animals may not be available, they could be treated as hyperparameters of the model. Alternatively, simulated or synthetic data (which may adhere more or less closely to the assumptions in the proposed model) could be used to evaluate in which conditions the models are accurate. Without any such evaluation, though, it is difficult to tell if the models are meaningful.

Similarly, although the control measures were introduced and associated with a cost, the optimal policy was not really discussed. Additionally, the costs presumably give rise to a Pareto front of multiple possible policies with different tradeoffs among the different control measures. In principle, such tradeoffs could be used to guide policymakers. However, these were not at all discussed in the paper.

Finally, it is not clear to what extent the proposed model is specific to monkeypox or whether it would be applicable to most (or even all) other zoonotic diseases. Elaborating on this could help broaden the impact of this work.

Clarity

The paper has numerous grammatical mistakes. I do not believe they affect understanding the paper, but they do become distracting. The paper needs another round of editing.

The references are not consistently formatted.

Otherwise, the paper is generally clear, and the various model parameters are well-described. Some of the lengthy equations could likely be simplified or moved to an appendix.

Reproducibility

The submission does not include any supplementary code or data. Nevertheless, I believe results similar to those in the paper could be reproduced based on the given equations.

**Strength And Weaknesses:**

The main strength of the paper lies in the novelty of explicitly modeling the cross-transmission between human and animal populations.

The main weaknesses of the work are in the limited methodological contributions as well as lack of data-driven evaluation of the model. As described in more detail below, the evaluation could be improved by either including some type of real-world data for validating the model or at least providing a more thorough qualitative analysis (supported by synthetic/simulated data).

**Summary Of The Paper:**

In this work, the authors develop a differential equation-based model of the dynamics for monkeypox. The model explicitly includes both human and animal populations, as well as the cross-transmission from animal to human. The authors also propose several control measures affecting various transition rates in the model; a control theory approach is then used to derive optimal policies for the control measures. Finally, a limited set of numerical simulations confirm the derived analytic results concerning the plausibility of the model.

**Summary Of The Review:**

Overall, while the paper proposes some interesting approaches to modeling monkeypox and some associated control strategies, the limited methodological contribution and complete lack of data-driven analysis likely limits the interest at a venue like ICLR. Improved discussion of the resulting policy for control measures, as well as a general discussion of the use of the approach beyond just monkeypox, may improve the impact.

---

### Official Review · Reviewer_zkQd · 2022-11-02

**Confidence:** 4
**Clarity, Quality, Novelty And Reproducibility:** look above
**Correctness:** 3
**Technical Novelty And Significance:** 2
**Empirical Novelty And Significance:** Not applicable
**Recommendation:** 1

**Details Of Ethics Concerns:**

Please look at my summary of the paper review

**Strength And Weaknesses:**

look above

**Summary Of The Paper:**

This paper introduces a compartmental model which allows for cross infections between animals and humans. The model is used to study the dynamics of the Monkeypox spread.

I have major concerns for this paper to be almost a double submission of the paper http://www.aimspress.com/article/doi/10.3934/mbe.2022633  (Stochastic modeling of the Monkeypox 2022 epidemic with cross-infection hypothesis in a highly disturbed environment). Now it could be a case that it is not but then there are too many similarities between both papers
* Abstract reads almost the same
* Introduction is more or less same (in both points I am talking about literal word to word copies).
* Proposed models are same if you account for description in the appendix
* Even the major listed contributions are same for both papers
* Figure 1 is more or less identical.

Now a case can be made that first paper is essentially the model and this second paper is mainly about the optimal control. However, I doubt that as even in first paper authors talk about scenarios which are akin to the optimal control setting here and importantly in words of authors itself, they list their main contributions as:-
```
"
   * The cross infection between humans and animals plays a significant role in the dynamics
     of monkeypox virus transmission. We, therefore, propose a model based on the hypothesis
     of cross-infection between humans and animals. The model has two blocks: humans and
      animals.
   • The first block describes the evolution of monkeypox in the human population, while the
      second block represents the evolution of the monkeypox virus among animals.
   • Four time-dependent control measures are then introduced in the model to demonstrate
      the utilization of optimal control measures: to minimize the infectious individuals and
      maximize the recovered population. Particularly, reducing the risk of disease transmission
       by educating people to rise awareness of risk factors, treatment of infected individuals, and
       restrictions on animals.
"
```

Now first two can't be contributions of this paper in presence of the first paper I am talking about. Also quite mysteriously the first paper is nowhere cited in the main paper which raises even more concern.

I will finish my review here and hope that authors prove me wrong and their rebuttal makes me look at paper differently to evaluate other points.

**Summary Of The Review:**

look above

---

### Decision · Program_Chairs · 2023-01-20

**Decision:**

Reject

**Justification For Why Not Higher Score:**

The paper is very similar both in terms of content and actual text to a published paper not cited by the authors.

**Justification For Why Not Lower Score:**

N/A

**Metareview: Summary, Strengths And Weaknesses:**

This paper develops a differential equation-based model of the dynamics for monkeypox which allows for cross infections between animals and humans. All the reviewers have a variety of concerns, most prominently, regarding the lack of novelty in light of the related work. It is  particularly concerning that one published paper not cited by the authors was significantly similar both in terms of the actual contribution and textual similarity. The authors did not engage with the reviewers during the rebuttal period.